# ADVERSARIAL AUTOMIXUP

**Huafeng Qin**[1,*,†]  **Xin Jin**[1,*]  **Yun Jiang**[1]  **Mounim A. El-Yacoubi**[2]  **Xinbo Gao**[3]
[1]Chongqing Technology and Business University
[2]Telecom SudParis, Institut Polytechnique de Paris
[3]Chongqing University of Posts and Telecommunications
[*] Equal contribution   [†] Corresponding author

## ABSTRACT

Data mixing augmentation has been widely applied to improve the generalization ability of deep neural networks. Recently, offline data mixing augmentation, *e.g.* handcrafted and saliency information-based mixup, has been gradually replaced by automatic mixing approaches. Through minimizing two sub-tasks, namely, mixed sample generation and mixup classification in an end-to-end way, AutoMix significantly improves accuracy on image classification tasks. However, as the optimization objective is consistent for the two sub-tasks, this approach is prone to generating consistent instead of diverse mixed samples, which results in overfitting for target task training. In this paper, we propose AdAutomixup, an adversarial automatic mixup augmentation approach that generates challenging samples to train a robust classifier for image classification, by alternatively optimizing the classifier and the mixup sample generator. AdAutomixup comprises two modules, a mixed example generator, and a target classifier. The mixed sample generator aims to produce hard mixed examples to challenge the target classifier, while the target classifier's aim is to learn robust features from hard mixed examples to improve generalization. To prevent the collapse of the inherent meanings of images, we further introduce an exponential moving average (EMA) teacher and cosine similarity to train AdAutomixup in an end-to-end way. Extensive experiments on seven image benchmarks consistently prove that our approach outperforms the state of the art in various classification scenarios. The source code is available at https://github.com/JinXins/Adversarial-AutoMixup.

## 1 INTRODUCTION

Due to their robust feature representation capacity, Deep neural network models, such as convolutional neural networks (CNN) and transformers, have been successfully applied in various tasks, *e.g.*, image classification (Li et al., 2022c; Krizhevsky et al., 2012; Li et al., 2022a; 2024), object detection (Bochkovskiy et al., 2020), and natural language processing (Vaswani et al., 2017). One of the important reasons is that they generally exploit large training datasets to train massive network parameters. When the data is insufficient, however, they become prone to over-fitting and make overconfident predictions, which may degrade the generalization performance on test examples.

To alleviate these drawbacks, data augmentation (DA) is proposed to generate samples to improve generalization on downstream target tasks. $Mixup$ (Zhang et al., 2017), a recent DA scheme, has received increasing attention as it can produce virtual mixup examples via a simple convex combination of pairs of examples and their labels to effectively train a deep learning (DL) model. DA approaches (Li et al., 2021; Shorten & Khoshgoftaar, 2019; Cubuk et al., 2018; 2020; Fang et al., 2020; Ren et al., 2015; Li et al., 2020), proposed for image classification, can be broadly split into three categories: 1) Handcrafted-based mixup augmentation approaches, where patches from one image are randomly cut and pasted onto another. The ground truth label of the latter is mixed with the label of the former proportionally to the area of the replaced patches. Representative approaches include CutMix (Yun et al., 2019), Cutout (DeVries & Taylor, 2017), ManifoldMixup (Verma et al., 2019), and ResizeMix (Qin et al., 2020). CutMix and ResizeMix, as shown in Fig. 1, generate mixup samples by randomly replacing a patch in an image with patches from another; 2) Saliency-guided mixup augmentation approaches that generate, based on image saliency maps, high-quality samples by preserving regions of maximum saliency. Representative approaches (Uddin et al., 2020; Walawalkar et al., 2020; Kim et al., 2020; Park et al., 2021; Liu et al., 2022c) learn the optimal

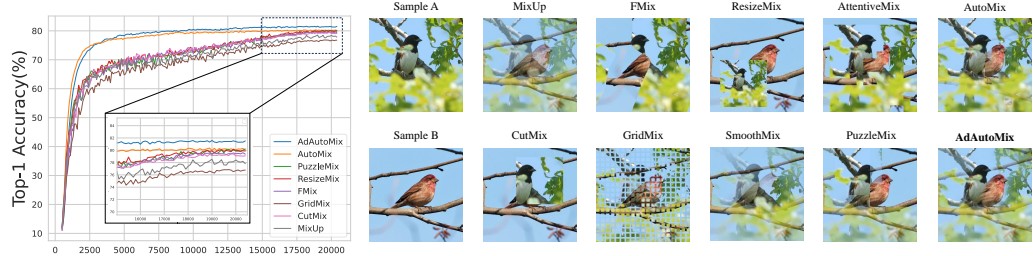

Figure 1: Mixed images of various approaches. (a) Accuracy of ResNet18 trained by different mixup approaches with 200 epochs on CIFAR100. (b) Mixed images of various mixup-based approaches.

mixing policy by maximizing the saliency regions; 3) Automatic Mixup-based augmentation approaches, that learn a model, *e.g.* DL model, instead of a policy, to automatically generate mixed images. (Liu et al., 2022d) for example, proposed an AutoMix model for DA, consisting of a target classifier and a generative network, to automatically generate mixed samples to train a robust classifier by alternatively optimizing the target classifier and the generative network.

The handcrafted mixup augmentation approaches, however, randomly mix images without considering their contexts and labels. The target objects, therefore, may be missed in the mixed images, resulting in a label mismatch problem. Saliency-guided-based mixup augmentation methods can alleviate the problem as the images are combined with supervising information, namely the maximum saliency region. These mixup models, related to the first two categories above, share the same learning paradigm: an augmented training dataset generated by random or learnable mixing policy and a DL model for image classification. As image generation is not directly related to the target task, *i.e.*, classification, the generated images guided by human prior knowledge, *i.e.*, saliency-based, may not be effective for target network training. Moreover, it is impossible to generate all possible mixed instances for target training. The randomly selected synthesized samples thus may not be representative of the classification task, ultimately degrading classifier generalization. Besides, such generated samples will be input to the target network repeatedly, resulting in inevitable overfitting over long epoch training. To overcome these problems, automatic mixup-based augmentation approaches generate augmented images by a sub-network with a good complexity-accuracy trade-off. This approach comprises two sub-tasks: a mixed sample generation module and a classification module, conjointly optimized by minimizing the classification loss in an end-to-end way. As the optimizing goal is consistent for the two sub-tasks, however, the generation module may not be effectively guided and may produce, consequently, simple mixed samples to achieve such a goal, which limits sample diversification. The classifier trained on such simple examples is prone, therefore, to suffer from overfitting, leading to poor generalization performance on the testing set. Another limitation is that current automatic mixup approaches mix two images only for image generation, where the rich and discriminating information is not efficiently exploited.

To solve these problems, we propose in this paper *AdAutomixup*, an adversarial automatic mixup augmentation approach to automatically generate mixed samples with adversarial training in an end-to-end way, as shown in Fig. 2. First, an attention-based generator is investigated to dynamically learn discriminating pixels from a sample pair associated with the corresponding mixed labels. Second, we combine the attention-based generator with the target classifier to build an adversarial network, where the generator and the classifier are alternatively updated by adversarial training. Unlike AutoMix (Liu et al., 2022d), a generator is learned to increase the training loss of the target network through generating adversarial samples, while the classifier learns more robust features from hard examples to improve generalization. Furthermore, any set of images, instead of two images only, can be taken as an input to our generator for mixing image generation, which results in more diversification of the mixed samples. Our main contributions are summarized as follows.

(a) We present an online data mixing approach based on an adversarial learning policy, trained end-to-end to automatically produce mixed samples.

(b) We propose an adversarial framework to jointly optimize the target network training and the mixup sample generator. The generator aims to produce hard samples to increase the target network loss while the target network, trained on such hard samples, learns a robust representation to improve classification. To avoid the collapse of the inherent meanings of images, we apply an exponential moving average (EMA) and cosine similarity to reduce the search space.

(c) We explore an attention-based mix sample generator that can combine multiple samples instead of only two samples to generate mixed samples. The generator is flexible as its architecture is not changed with the increase of input images.

## 2 RELATED WORK

**Hand-crafted based mixup augumentaion** Mixup (Zhang et al., 2017), the first hybrid data augmentation method, generates mixed samples by subtracting any two samples and their one-hot labels. ManifoldMixup (Verma et al., 2019) extended this mixup from input space to feature space. To exploit spatial locality, CutMix (Yun et al., 2019) crops out a region and replace it with a patch of another image. To improve MixUp and CutMix, FMix (Harris et al., 2020) uses random binary masks obtained by applying a threshold to low-frequency images sampled from the frequency space. RecursiveMix (Yang et al., 2022) iteratively resizes the input image patch from the previous iteration and pastes it into the current patch. To solve the strong "edge" problem caused by CutMix, SmoothMix (Jeong et al., 2021) blends mixed images based on soft edges, with the training labels computed accordingly.

**Saliency guided based mixup augmentation** SaliencyMix (Uddin et al., 2020), SnapMix (Huang et al., 2020) and Attentive-CutMix (Walawalkar et al., 2020) generate mixed images based on the salient region detected by the Class Activation Mapping(CAM) (Selvaraju et al., 2019) or saliency detector. Similarly, PuzzleMix (Kim et al., 2020) and Co-Mixup (Kim et al., 2021) propose an optimization strategy to obtain the optimal mask by maximizing the sample saliency region. These approaches, however, suffer from a lack of sample diversification as they always deterministically select regions with maximum saliency. To solve the problem, Saliency Grafting (Park et al., 2021) scales and thresholds the saliency map to grant all salient regions are considered to increase sample diversity. Inspired by the success of Vit (Dosovitskiy et al., 2021; Liu et al., 2021) in computer vision, adaptive mixing policies based on attentive maps, *e.g.*, TransMix (Chen et al., 2021), TokenMix (Liu et al., 2022a), TokenMixup (Choi et al., 2022), MixPro (Zhao et al., 2023), and SMMix (Chen et al., 2022), were proposed to generate mixed images.

**Automatic Mixup based augmentation** Mixup approaches in the first two categories allow a trade-off between precise mixing policies and optimization complexity, as the image mixing task is not directly related to the target classification task during the training process. To solve this problem, AutoMix (Liu et al., 2022d) divides the mixup classification into two sub-tasks, mixed sample generation and mixup classification, and proposes an automatic mixup framework where the two sub-tasks are optimized jointly, instead of independently. During training, the generator continuously produces the mixed samples while the target classifier is preserved for classification. In recent years, adversarial data augmentation (Zhao et al., 2020) and generative adversarial networks (Antoniou et al., 2017) were proposed to automatically generate images for data augmentation. To solve the domain shift problem, Adversarial MixUp (Zhang et al., 2023; Xu et al., 2019) have been investigated to synthesize mix samples or features for domain adaptation. Although there are very few works for automatic mixup, it will become a research trend in the future.

## 3 ADAUTOMIX

In this section, we present the implementation of AdAutoMix, which is composed of a target classifier and a generator, as shown in Fig. 2. First, we introduce the mixup classification problem and define the loss functions. Then, we detail our attention-based generator that learns dynamically the augmentation mask policy for image generation. Finally, we show how the target classifier and the generator are jointly optimized in an end-to-end way.

### 3.1 DEEP LEARNING-BASED CLASSIFIERS

Assume that $\mathbb{S} = \{x_s | s = 1, 2, ..., S\}$ is a training set, where $S$ is the number of the images. We select any $N$ samples from $\mathbb{S}$ to obtain a sample set $\mathbb{X} = \{x_1, x_2, ..., x_N\}$, with $\mathbb{Y} = \{y_1, y_2, ..., y_N\}$ its corresponding label set. Let $\psi_W$ be any feature extraction model, *e.g.*, ResNet (He et al., 2016), where $W$ is a trainable weight vector. The classifier maps example $x \in \mathbb{X}$ into label $y \in \mathbb{Y}$. A DL classifier $\psi_W$ is implemented to predict the posterior class probability, and $W$ are learned by minimizing the classification loss, *i.e.* the cross entropy (CE) loss in Eq.(1):

$$L_{ce}(\psi_W, y) = -y\log(\psi_W(x)). \tag{1}$$

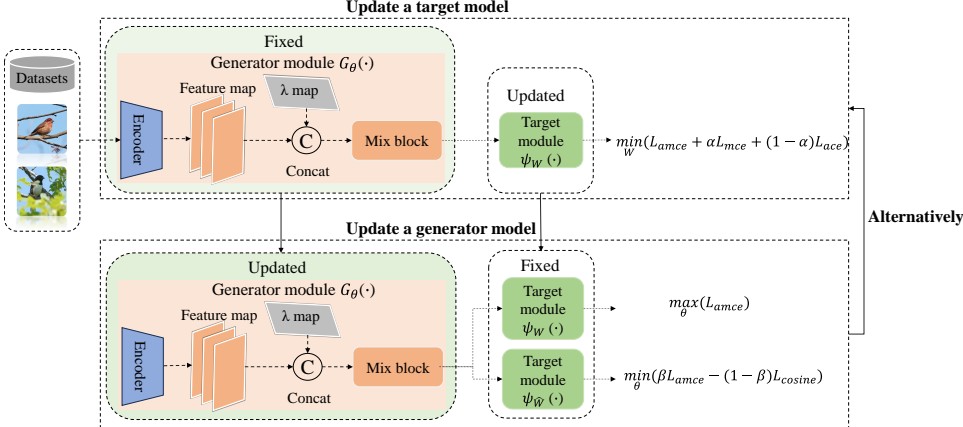

Figure 2: Illustration of AdAutoMix framework. AdAutoMix consists of a generator module and a target module, which are alternatively trained end-to-end. The generator module aims to produce hard samples to challenge the target network while the target network, trained on such hard samples, learns a robust feature representation for classification.

For $N$ samples in sample set $\mathbb{X}$, the average cross-entropy (ACE) loss is computed by Eq.(2):

$$L_{ace}(\psi_W, \mathbb{Y}) = \sum_{n=1}^{N} (L_{ce}(\psi_W(x_n), y_n) * \lambda_n). \tag{2}$$

where $*$ is the scalar multiplication. In the mixup classification task, we input any $N$ images associated with mixed ratios $\lambda$ to a generator $G_\theta(\cdot)$ that outputs a mixed sample $x_{mix}$, as defined in Eq.(8) from section 3.2. Similarly, the label for such a mixed image $x_{mix}$ is obtained by $y_{mix} = \sum_{n=1}^{N} y_n \odot \lambda_n$. $\psi_W$ is optimized by average mixup cross-entropy (AMCE) loss in Eq.(3):

$$L_{amce}(\psi_W, \mathbb{Y}) = L_{ce}(\psi_W(x_{mix}), y_{mix}). \tag{3}$$

Similarly, we also compute the mixup cross-entropy (MCE) by Eq.(4):

$$L_{mce}(\psi_W, y_{mix}) = L_{ce}(\psi_W(\sum_{n=1}^{N} (x_n * \lambda_n)), y_{mix}). \tag{4}$$

### 3.2 GENERATOR

As described in Section 2, most existing approaches mix two samples by manually designed policies or automatic learning policies, which results in insufficient exploitation of the supervised information that might be provided by the training samples for data augmentation. In our work, we present a universal generation framework to extend the two-image mixing to multiple-image mixing. To learn a robust mixing policy matrix, we leverage a self-attention mechanism to propose an attention-based mixed sample generator, as shown in Fig. 3. As described in Section 3.1, $\mathbb{X} = \{x_n | n = 1, 2, ..., N\}$ is a sample set with $N$ original training samples and $\mathbb{Y} = \{Y_n | n = 1, 2, ..., N\}$ are the corresponding labels. We define $\lambda = \{\lambda_1, \lambda_2, ..., \lambda_N\}$ as the mixed ratio set for the images with their sum constrained to be equal to 1. As shown in Fig. 3, each image in an image set is first mapped to a feature map with encoder $E_\phi$, which is updated by an exponential moving average of the target classifier, $i.e. \widehat{\phi} = \xi\widehat{\phi} + (1 - \xi)W'$, where $W'$ is the partial weight of the target classifier. In our experiments, existing classifiers, ResNet18, ResNet34, and ResNeXt50, are used as target classifiers, and $W'$ is the weight vector of the first three layers in the target classifier. Then, the mixed ratios are embedded into the resulting feature map to enable the generator to learn mask policies for image mixing. For example, given $n$th image $x_n \in R^{W \times H}$, where $W$ and $H$ represent image size, we input it to an encoder and take outputs from its $l$th layer as feature map $z_n^l \in R^{C \times w \times h}$, where $C$ is the number of channels, and $w$ and $h$ represent map size . Then, we build a matrix with size $w \times h$ with all values equal to 1, multiplied by the corresponding ratio $\lambda_n$ to obtain embedding matrix $M_{\lambda_n}$. We embed $\lambda_n$ with the $l$th feature map in a simple and efficient way by concatenating $z_{\lambda_n}^l = concat(M_{\lambda_n}, z_n^l) \in R^{(C+1) \times w \times h}$. The embedding feature map $z_{\lambda_n}^l$ is mapped to three embedding vectors by three CNNs with $1 \times 1$ kernel (as shown in Fig. 3), respectively. Therefore, we obtain three vectors $q_n$, $k_n$, and $v_n$ for the $n$th image $x_n$. Note that the channel number is reduced

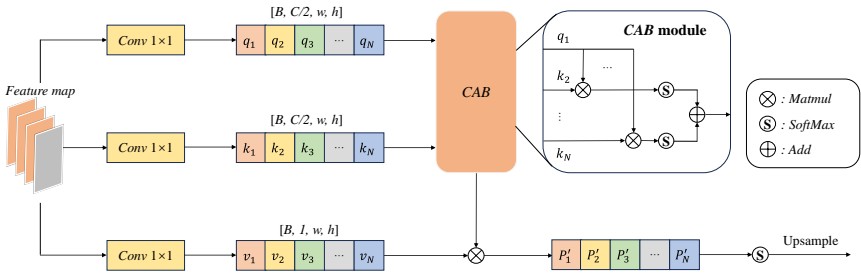

Figure 3: Mixed module: the cross attention block (CAB), used to learn the policy matrix for each image, is combined with $v_i(1 = 1, 2, ..., N)$ values to compute the policy matrix for image mixing.

to its half for $q_n$ and $k_n$ to save computation time and is set to 1 for $v_n$. In this way, the embedding vectors of all images are computed and denoted by $q_1, q_2,...,q_N$, $k_1, k_2,...,k_N$, and $v_1, v_2,...,v_N$. The cross attention block (CAB) (as shown in Fig. 3) for the $n$th image is computed by Eq. (5):

$$P_n = Softmax\left(\frac{\sum_{i=1,i\neq n}^{N} q_n^T k_i}{\sqrt{d}}\right) v_n,$$ (5)

where $d$ is the normalization term. We concatenate $N$ attention matrices by Eq. (6):

$$P = Softmax(Concat(P_1, P_2, ..., P_N)).$$ (6)

The matrix $P \in R^{N \times wh \times wh}$ is resized to $P' \in R^{N \times W \times H}$ by upsampling. We split $N$ matrices, $P'_1, P'_2, ..., P'_N$ from $P'$, treated as mask policy matrices to mix images in the sample set $\mathbb{X}$ by Eq.(7):

$$x_{mix} = \sum_{n=1}^{N} x_n \odot P'_n,$$ (7)

where $\odot$ denotes the Hadamard product. To facilitate representation, the mixed image generation procedure is denoted as a generator $G_\theta$ by Eq.(8):

$$x_{mix} = G_\theta(\mathbb{X}, \lambda),$$ (8)

where $\theta$ represents all the learnable parameters of the generator.

## 3.3 ADVERSARIAL AUGMENTATION

This section provides the adversarial framework we propose to jointly optimize the target network $\psi_W$ and the generator $G_\theta$ through adversarial learning. Concretely, the generator $G_\theta$ attempts to produce an augmented mixed image set to increase the loss of target network $\psi_W$ while target network $\psi_W$ aims to minimize the classification loss. An equilibrium can be reached where the learned representation reaches maximized performance.

### 3.3.1 ADVERSARIAL LOSS

As shown in Eq.(8), the generator takes $\mathbb{X}$ and the set of mixed ratio $\lambda$ as input and outputs a synthesized image $x_{mix}$ to challenge the target classifier. The latter receives either a real or a synthesized image from the generator as its input and then predicts its probability of belonging to each class. The adversarial loss is defined by the following minimax problem to train both players by Eq.(9):

$$W^*, \theta^* = \underset{W}{\arg\min}\underset{\theta}{\max}\left[\underset{\mathbb{X}\in\mathbb{S}}{\mathbb{E}} \left[\text{L}_{\text{amce}}(\psi_W, \mathbb{Y})\right]\right],$$ (9)

where $\mathbb{S}$ and $\mathbb{X}$ are the training set and image set, respectively. A robust classifier should correctly classify not only the mixed images, but also the original ones, so we incorporate two regularization terms $\text{L}_{\text{mce}}(\psi_W(x_{mix}, y_{mix}))$ and $\text{L}_{\text{ace}}(\psi_W, \mathbb{Y})$ to enhance performance. Accordingly, the objective function is rewritten as shown by Eq.(10):

$$W^*, \theta^* = \underset{W}{\arg\min}\underset{\theta}{\max}\left[\underset{\mathbb{X}\in\mathbb{S}}{\mathbb{E}} \left[\text{L}_{\text{amce}}(\psi_W, \mathbb{Y}) + \alpha\text{L}_{\text{mce}}(\psi_W, y_{mix}) + (1 - \alpha)\text{L}_{\text{ace}}(\psi_W, \mathbb{Y})\right]\right].$$ (10)

To optimize parameter $\theta$, $G_\theta(\cdot)$ produces images with given image sets to challenge the classifier. It is possible, therefore, that the inherent meanings of images (*i.e.* their semantic meaning) collapse.

To tackle this issue, we introduce cosine similarity and a teacher model as two regularization terms to control the quality of mixed images. The loss is changed accordingly, as shown by Eq.(11):

$$W^*, \theta^* = \arg\min_W \max_\theta [\mathop{\mathbb{E}}_{\mathbb{X} \in \mathbb{S}} [\mathrm{L}_{\mathrm{amce}}(\psi_W, \mathbb{Y}) + \alpha \mathrm{L}_{\mathrm{mce}}(\psi_W, y_{mix}) + (1-\alpha)\mathrm{L}_{\mathrm{ace}}(\psi_W, \mathbb{Y})$$
$$- \beta \mathrm{L}_{\mathrm{amce}}(\psi_{\widehat{W}}, \mathbb{Y}) + (1-\beta)\mathrm{L}_{\mathrm{cosine}}]], \quad (11)$$

where $\mathrm{L}_{\mathrm{cosine}} = \sum_{n=1}^{N} cosine(\psi_{\widehat{W}}(x_{mix}), \psi_{\widehat{W}}(x_n)) * \lambda_n$, $cosine(\cdot)$ is cosine similarity function, and $\psi_{\widehat{W}}$ is a teacher model whose weights are updated as an exponential moving average of the target (EMA) models weights, *i.e.* $\widehat{W} \leftarrow \xi\widehat{W} + (1-\xi)W$. Notice that $\mathrm{L}_{\mathrm{ce}}(\psi_W, y)$ is the standard cross-entropy loss. $\mathrm{L}_{\mathrm{ace}}(\psi_W, \mathbb{Y})$ loss facilitates the backbone to provide a stable feature map at early stage so that it speeds up convergence. Target loss $\mathrm{L}_{\mathrm{amce}}(\psi_W, \mathbb{Y})$ aims to learn task-relevant information in the generated mixed samples. $\mathrm{L}_{\mathrm{mce}}(\psi_W, y_{mix})$ facilitates the capture of task-relevant information in the original mixed samples. $\mathrm{L}_{\mathrm{cosine}}$ and $\mathrm{L}_{\mathrm{amce}}(\psi_{\widehat{W}}, \mathbb{Y})$ are used to control the quality of generation mix images.

## 3.4 ADVERSARIAL OPTIMIZATION

Similarly to many existing adversarial training algorithms, it is hard to directly find a saddle point $(W^*, \theta^*)$ solution to the minimax problem in Eq.(11). Alternatively, a pair of gradient descent and ascent are employed to update the target network and the generator.

Consider target classifier $\psi_W(\cdot)$ with a loss function $\mathrm{L}_{\mathrm{ce}}(\cdot)$, where the trained generator $G_\theta(\cdot)$ maps multiple original samples to a mixed sample. The learning process of the target network can be defined as the minimization problem in Eq.(12):

$$W^* = \arg\min_W [\mathop{\mathbb{E}}_{\mathbb{X} \in \mathbb{S}} [\mathrm{L}_{\mathrm{amce}}(\psi_W, \mathbb{Y}) + \alpha \mathrm{L}_{\mathrm{mce}}(\psi_W, y_{mix}) + (1-\alpha)\mathrm{L}_{\mathrm{ace}}(\psi_W, \mathbb{Y})$$
$$- \beta \mathrm{L}_{\mathrm{amce}}(\psi_{\widehat{W}}, \mathbb{Y}) + (1-\beta)\mathrm{L}_{\mathrm{cosine}}]]. \quad (12)$$

The problem in Eq. (12) is usually solved by vanilla SGD with a learning rate of $\delta$ and a batch size of $B$, and the training procedure for each batch can be computed by Eq.(13):

$$W(t+1) = W(t) - \delta \nabla_W \frac{1}{K} \sum_{k=1}^{K} [\mathrm{L}_{\mathrm{amce}}(\psi_W, \mathbb{Y}) + \alpha \mathrm{L}_{\mathrm{mce}}(\psi_W, y_{mix}) + (1-\alpha)\mathrm{L}_{\mathrm{ace}}(\psi_W, \mathbb{Y})$$
$$- \beta \mathrm{L}_{\mathrm{amce}}(\psi_{\widehat{W}}, \mathbb{Y}) + (1-\beta)\mathrm{L}_{\mathrm{cosine}}]. \quad (13)$$

where $K$ is the number of mixed images or image sets produced from patch set $B$. As the cosine similarity and the teacher model are independent of $W$, Eq.(13) can be rewritten as Eq.(14):

$$W(t+1) = W(t) - \delta \nabla_W \frac{1}{K} \sum_{k=1}^{K} [\mathrm{L}_{\mathrm{amce}}(\psi_W, \mathbb{Y}) + \alpha \mathrm{L}_{\mathrm{mce}}(\psi_W, y_{mix}) + (1-\alpha)\mathrm{L}_{\mathrm{ace}}(\psi_W, \mathbb{Y})]. \quad (14)$$

Note that the training procedure can be regarded as an average over $K$ instances of gradient computation, which can reduce gradient variance and accelerate the convergence of the target network. However, training may suffer easily from over-fitting due to the limited training data over a long training epoch. To overcome this problem, different from AutoMix (Liu et al., 2022d), our mixup augmentation generator generates a set of harder mixed samples to increase the loss of the target classifier, which results in a minimax problem to self-train the network. Such a self-supervised objective may be sufficiently challenging to prevent the target classifier from overfitting the objective. Therefore, the objective is defined as the following maximization problem in Eq.(15):

$$\theta^* = \arg\max_\theta [\mathop{\mathbb{E}}_{\mathbb{X} \in \mathbb{S}} [\mathrm{L}_{\mathrm{amce}}(\psi_W, \mathbb{Y}) - \beta \mathrm{L}_{\mathrm{amce}}(\psi_{\widehat{W}}, \mathbb{Y}) + (1-\beta)\mathrm{L}_{\mathrm{cosine}}]]. \quad (15)$$

To solve the above problem, we employ a gradient ascent to update the parameter with a learning rate of $\gamma$, which is defined in Eq.(16):

$$\theta(t+1) = \theta(t) + \gamma \nabla_W \frac{1}{K} \sum_{k=1}^{K} [\mathrm{L}_{\mathrm{amce}}(\psi_W, \mathbb{Y}) - \beta \mathrm{L}_{\mathrm{amce}}(\psi_{\widehat{W}}, \mathbb{Y}) + (1-\beta)\mathrm{L}_{\mathrm{cosine}}]. \quad (16)$$

Intuitively, the optimization of Eq.(16) is the combination of two sub-tasks, the maximization of $\mathrm{L}_{\mathrm{ce}}(\psi_W(x_{mix}, y_{mix}))$ and the minimization of $\beta \mathrm{L}_{\mathrm{amce}}(\psi_{\widehat{W}}, \mathbb{Y}) - (1-\beta)\mathrm{L}_{\mathrm{cosine}}$. This tends to push

the synthesized mixed samples far away from the real samples to increase diversity, while ensuring the synthesized mixed samples are recognizable for a teacher model and kept, within a constraint similarity to the feature representation of original images, so as to avoid collapsing the inherent meanings of images. This scheme enables generating challenging samples by closely tracking the updates of the classifier. We provide some mixed samples in Appendix B.2 and B.3.

## 4 EXPERIMENTS

To estimate our approach performance, we conducted extensive experiments on seven classification benchmarks, namely CIFAR100 (Krizhevsky et al., 2009), Tiny-ImageNet (Chrabaszcz et al., 2017), ImageNet-1K (Krizhevsky et al., 2012), CUB-200 (Wah et al., 2011), FGVC-Aircraft (Maji et al., 2013) and Standford-Cars (Krause et al., 2013) (Appendix A.1). For fair assessment, we compare our AdAutoMixup with some current Mixup methods, *i.e.* Mixup (Zhang et al., 2017), CutMix (Yun et al., 2019), ManifoldMix (Verma et al., 2019), FMix (Harris et al., 2020), ResizeMix (Qin et al., 2020), SaliencyMix (Uddin et al., 2020), PuzzleMix (Kim et al., 2020) and AutoMix (Liu et al., 2022d). To verify our approach generalizability, five baseline networks, namely ResNet18, ResNet34, ResNet50 (He et al., 2016), ResNeXt50 (Xie et al., 2017), Swin-Transformer (Liu et al., 2021) and ConvNeXt(Liu et al., 2022b), are used to compute classification accuracy. We have implemented our algorithm on the open-source library OpenMixup (Li et al., 2022b). Some common parameters follow the experimental settings of AutoMix and we provide our own hyperparameters in Appendix A.2. For all classification results, we report the mean performance of 3 trials where the median of top-1 test accuracy in the last 10 training epochs is recorded for each trial. To facilitate comparison, we mark the best and second best results in bold and cyan.

### 4.1 CLASSIFICATION RESULTS

#### 4.1.1 DATASET CLASSIFICATION

We first train ResNet18 and ResNeXt50 on CIFAR100 for 800 epochs, using the following experimental setting: The basic learning rate is 0.1, dynamically adjusted by cosine scheduler, SGD (Loshchilov & Hutter, 2016) optimizer with momentum of 0.9, weight decay of 0.0001, batch size of 100. To train ViT-based models, *e.g.* Swin-Tiny Transformer and ConvNeXt-Tiny, we train them with AdamW (Loshchilov & Hutter, 2019) optimizer with weight decay of 0.05, batch size of 100, 200 epochs. On Tiny-ImageNet, except for a learning rate of 0.2 and training over 400 epochs, training settings are similar to the ones used in CIFAR100. On ImageNet-1K, we train ResNet18, ResNet34 and ResNet50 for 100 epochs using PyTorch-style setting . The experiments implementation details are provided in Appendix A.3

Table 1 and Fig. 1 show that our method outperforms the existing approaches on CIFAR100. After training by our approach, ResNet18 and ResNeXt50 achieve an accuracy improvement of **0.28**% and **0.58**% w.r.t the second best results, respectively. Similarly, ViT-based approaches achieve the highest classification accuracy of 84.33 % and 83.54% and outperform the previous best approaches with an improvement of **1.66**% and **0.24**%. On the Tiny-ImageNet datasets, our AdAutoMix consistently outperforms existing approaches in terms of improving the classification performance of ResNet18 and ResNeXt50, *i.e.* **1.86** % and **2.17**% significant improvement w.r.t the second best approaches. Also, Table 1 shows that AdAutoMix achieves an accuracy improvement (0.36% for ResNet18, 0.3% for ResNet34, and 0.13% ResNet50) on the ImageNet-1K large scale dataset.

Table 1: Top-1 accuracy (%)↑ of mixup methods on CIFAR-100, Tiny-ImageNet and ImageNet-1K.

| Method | CIFAR100 | | CIFAR100 | | Tiny-ImageNet | | ImageNet-1K | | |
|---|---|---|---|---|---|---|---|---|---|
| | ResNet18 | ResNeXt50 | Swin-T | ConvNeXt-T | ResNet18 | ResNeXt50 | ResNet18 | ResNet34 | ResNet50 |
| Vanilla | 78.04 | 81.09 | 78.41 | 78.70 | 61.68 | 65.04 | 70.04 | 73.85 | 76.83 |
| MixUp | 79.12 | 82.10 | 76.78 | 81.13 | 63.86 | 66.36 | 69.98 | 73.97 | 77.12 |
| CutMix | 78.17 | 81.67 | 80.64 | 82.46 | 65.53 | 66.47 | 68.95 | 73.58 | 77.17 |
| SaliencyMix | 79.12 | 81.53 | 80.40 | 82.82 | 64.60 | 66.55 | 69.16 | 73.56 | 77.14 |
| FMix | 79.69 | 81.90 | 80.72 | 81.79 | 63.47 | 65.08 | 69.96 | 74.08 | 77.19 |
| PuzzleMix | 81.13 | 82.85 | 80.33 | 82.29 | 65.81 | 67.83 | 70.12 | 74.26 | 77.54 |
| ResizeMix | 80.01 | 81.82 | 80.16 | 82.53 | 63.74 | 65.87 | 69.50 | 73.88 | 77.42 |
| AutoMix | 82.04 | 83.64 | 82.67 | 83.30 | 67.33 | 70.72 | 70.50 | 74.52 | 77.91 |
| **AdAutoMix** | **82.32** | **84.22** | **84.33** | **83.54** | **69.19** | **72.89** | **70.86** | **74.82** | **78.04** |
| Gain | +0.28 | +0.58 | +1.66 | +0.24 | +1.86 | +2.17 | +0.36 | +0.30 | +0.13 |

Table 2: Accuracy (%)↑ of mixup approaches on CUB-200, FGVC-Aircrafts and Standford-Cars.

| Method | CUB-200 | | FGVC-Aircrafts | | Standford-Cars | |
|---|---|---|---|---|---|---|
| | ResNet18 | ResNet50 | ResNet18 | ResNeXt50 | ResNet18 | ResNeXt50 |
| Vanilla | 77.68 | 82.38 | 80.23 | 85.10 | 86.32 | 90.15 |
| MixUp | 78.39 | 82.98 | 79.52 | 85.18 | 86.27 | 90.81 |
| CutMix | 78.40 | 83.17 | 78.84 | 84.55 | 87.48 | 91.22 |
| ManifoldMix | 79.76 | 83.76 | 80.68 | 86.60 | 85.88 | 90.20 |
| SaliencyMix | 77.95 | 81.71 | 80.02 | 84.31 | 86.48 | 90.60 |
| FMix | 77.28 | 83.34 | 79.36 | 86.23 | 87.55 | 90.90 |
| PuzzleMix | 78.63 | 83.83 | 80.76 | 86.23 | 87.78 | 91.29 |
| ResizeMix | 78.50 | 83.41 | 78.10 | 84.08 | 88.17 | 91.36 |
| AutoMix | 79.87 | 83.88 | 81.37 | 86.72 | 88.89 | 91.38 |
| **AdAutoMix** | **80.88** | **84.57** | **81.73** | **87.16** | **89.19** | **91.59** |
| Gain | **+1.01** | **+0.69** | **+0.36** | **+0.44** | **+0.30** | **+0.21** |

### 4.1.2 FINE-GRAINED CLASSIFICATION

On CUB-200, FGVC-Aircrafts, and Standford-Cars, we fine-tune pretrained ResNet18, ResNet50, and ResNeXt50 using SGD optimizer with momentum of 0.9, weight decay of 0.0005, batch size of 16, 200 epochs, learning rate of 0.001, dynamically adjusted by cosine scheduler. The results in Table 2 show that AdAutoMix achieves the best performance and significantly improves the performance of vanilla (**3.20**%/**2.19**% on CUB-200, **1.5**%/**2.06**% on Aircraft and **2.87**%/**1.44**% on Cras), which implies that AdAutoMix is also robust to more challenging scenarios.

### 4.2 CALIBRATION

DNNs are prone to suffer from getting overconfident in classification tasks. Mixup methods can effectively alleviate this problem. To this end, we compute the expected calibration error (ECE) of various mixup approaches on the CIFAR100 dataset. It can be seen from the experimental results in Fig. 4 that our method achieves the lowest ECE, *i.e.* 3.2%, w.r.t existing approaches. We provide more experimental results in Table 6 in Appendix A.5

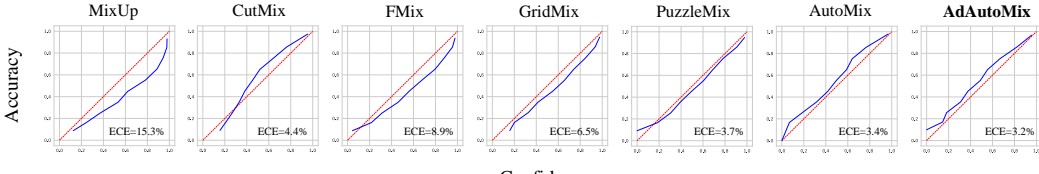

Figure 4: Calibration plots of Mixup variant on CIFAR100 using ResNet18.

### 4.3 ROBUSTNESS

We carried out experiments on CIFAR100-C (Hendrycks & Dietterich, 2019) to verify robustness against corruption. A corrupted dataset is manually generated to include 19 different corruption types (noise, blur, fog, brightness, etc.). We compare our AdAutoMix with some popular mixup algorithms: CutMix, FMix, PuzzleMix, and AutoMix. Table 4 shows that our approach achieves the highest recognition accuracy for both clean and corrupted data, *i.e.* 1.53% and 0.40% classification accuracy improvement w.r.t AutoMix. We further investigate robustness against the FGSM (Goodfellow et al., 2015) white box attack of 8/255 $\ell_\infty$ epsilon ball following (Zhang et al., 2017) . Our AdAutoMix significantly outperforms existing methods, as shown in Table 4.

### 4.4 OCCLUSION ROBUSTNESS

To analyze the AdAutoMix robustness against random occlusion (Naseer et al., 2021), we build image sets by randomly masking images from datasets CIFAR100 and CUB200 with 16×16 patches, using different mask ratios (0-100%). We input the resulting occluded images into two classifiers, Swin-Tiny Transformer and ResNet-50, trained by various Mixup models to compute test accuracy. From the results in Fig. 5 and in Table 7 in Appendix A.6, we observe that AdAutoMix achieves the highest accuracy with different occlusion ratios.

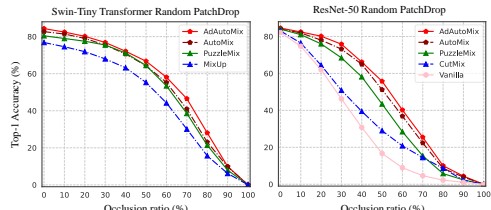

Figure 5: Robustness against image occlusion with different occlusion ratios.

Table 3: Top-1 accuracy (%)↑ with ResNet50 on CUB200 and Standford-Cars.

| Dataset | Vanilla | MixUp | CutMix | PuzzleMix | AutoMix | AdAutoMix |
|---------|---------|-------|--------|-----------|---------|-----------|
| CUB | 81.76 | 82.79 | 81.67 | 82.59 | 82.93 | **83.36(+0.43)** |
| Cars | 88.88 | 89.45 | 88.99 | 89.37 | 88.71 | **89.65(+0.20)** |

## 4.5 TRANSFER LEARNING

We further study the transferable abilities of the features learned by AdAutoMix for downstream classification tasks. The experimental settings in subsection 4.1.2 are used for transfer learning on CUB-200 and Standford-Cars, except for training now over 100 epochs. ResNet50 trained on ImageNet-1K is finetuned on CUB200 and Standford-Cars for classification. Table 3 shows that AdAutoMix achieves the best performance, which proves the efficacy of our approach for downstream tasks.

## 4.6 ABLATION EXPERIMENT

In AdAutoMix, four hyperparameters, namely the number of input images $N$, the weights $\alpha$, $\beta$, and mixed ratios $\lambda$, which are important to achieve high performance, are fixed in all experiments. To save time, we train the classifier on ResNet18 for 200 epochs by our AdAutoMixup. The accuracy of ResNet18 with different $\alpha$, $\beta$, $N$, and $\lambda$ are shown in Fig. 6 (a), (b), (c), and (d). Also, the classification accuracy of AdAutoMixup with different $\lambda$ and $N$ are depicted in Table. 9 and Table. 10 in Appendix A.8. AdAutoMix, with $N=3$, $\alpha =0.5$, $\beta=0.3$, and $\lambda =1$ as default, achieves the best performances on the various datasets. In addition, two regularization terms, $L_{mce}(\psi_W, y_{mix})$ and $L_{ace}(\psi_W, \mathbb{Y})$, attempt to improve

Table 4: Top-1 accuracy and FGSM error of ResNet18 with other methods.

| Method | Clean Acc(%)↑ | Corruption Acc(%)↑ | FGSM Error(%)↓ |
|--------|---------------|--------------------|-----------------|
| CutMix | 79.45 | 46.66 | 88.24 |
| FMix | 78.91 | 50.58 | 88.35 |
| PuzzleMix | 79.96 | 51.04 | 80.52 |
| AutoMix | 80.02 | 50.75 | 82.67 |
| AdAutoMix | **81.55** | **51.44** | **75.66** |

Table 5: Ablation experiments on CIFAR100 based on ResNet18 and ResNeXt50.

| Method | CIFAR100 | |
|--------|----------|----------|
| | ResNet18 | ResNeXt50 |
| Base($N = 3$) | 79.38 | 82.84 |
| $+0.5L_{mce} + 0.5L_{ace}$ | 80.04 | 84.12 |
| $-0.3L_{amce} + 0.7L_{cosine}$ | **81.55** | **84.40** |

classifier robustness, and another two regularization terms, namely cosine similarity $L_{cosine}$ and EMA model $L_{amce}(\psi_{\widehat{W}}, \mathbb{Y})$, aim to avoid the collapsing of the inherent meaning of images in AdAutoMix. We thus carry out experiments to evaluate the performance of each module concerning classifier performance improvement. To facilitate the description, we remove the four modules from AdAutoMix and denote the resulting approach as basic AdAutoMix. Then, we gradually incorporate the two modules $L_{mce}(\psi_W, \mathbb{Y})$ and $L_{ace}(\psi_W, \mathbb{Y})$, and the two modules $L_{amce}(\psi_{\widehat{W}}, \mathbb{Y})$ and $L_{cosine}$, and compute the classification accuracy. The experimental results in Table. 5 show that the $L_{mce}(\psi_W, y_{mix})$ and $L_{ace}(\psi_W, \mathbb{Y})$ improve classifier accuracy by about 0.66%. However, after incorporating $L_{amce}(\psi_{\widehat{W}}, \mathbb{Y})$ and $L_{cosine}$ to constraint the synthesized mixed images, we observe that the classification accuracy is significantly increased, namely 1.51% accuracy improvement, which implies that these two modules are capable of controlling the quality of generated images in the adversarial training. Also, we show the accuracy of our approach with gradually increasing individual regularization terms in Table. 8 in the Appendix. A.8. There is a similar trend that each regularization term improves accuracy.

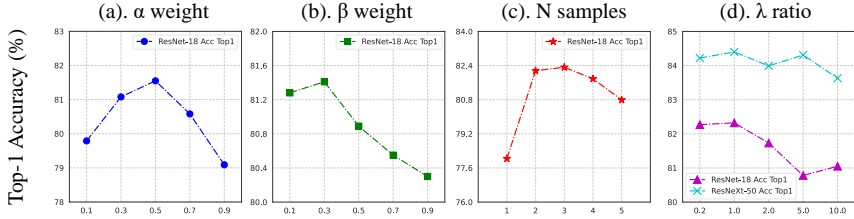

Figure 6: Ablation of hyperparameter $\alpha$, $\beta$, input samples and $\lambda$ of AdAutoMix on CIFAR100.

## 5 CONCLUSION

In this paper, we have proposed $AdAutoMixup$, a framework that jointly optimizes the target classifier and the mixing image generator in an adversarial way. Specifically, the generator produces hard mixed samples to increase the classification loss while the classifier is trained on the hard samples to improve generalization. In addition, the generator can handle multiple sample mixing cases. The experimental results on the six datasets demonstrate the efficacy of our approach.

ACKNOWLEDGEMENT

This work was supported in part by the Scientific Innovation 2030 Major Project for New Generation of AI under Grant 2020AAA0107300, in part by the National Natural Science Foundation of China (Grant No. 61976030), the Science Fund for Creative Research Groups of the Chongqing University (Grant No. CXQT21034), in part by the National Natural Science Foundation of China (Grant No. 62221005), in part by the National Natural Science Foundation of China (Grant No. U22A2096), in part by the Research on JY human-machine hybrid enhanced intelligence theory and method for command and decision-making (Grant No. 8091B012112), and in part by the Fund of Henan Provincial Science and Technology Department (Grant No. 222102210301). We thank all members who contribute to the OpenMixup community.

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

# A APPENDIX

## A.1 DATASET INFORMATION

We briefly introduce image datasets used in this paper. (1) CIFAR-100 (Krizhevsky et al., 2009) contains 50,000 training images and 10,000 test images in $32 \times 32$ resolutions, with 100 class settings. (2) Tiny-ImageNet (Chrabaszcz et al., 2017) contains 10,000 training images and 10,000 validation images of 200 classes in $64 \times 64$ resolutions. (3) ImageNet-1K (Krizhevsky et al., 2012) contains 1,281,167 training images and 50,000 validation images of 1000 classes. (4) CUB-200-2011 (Wah et al., 2011) contains 11,788 images from 200 wild bird species. FGVC-Aircrafts (Maji et al., 2013) contains 10,000 images of 100 classes of aricrafts and Standford-Cars (Krause et al., 2013) contains 8,144 training images and 8,041 test images of 196 classes.

## A.2 EXPERIMENTS HYPERPARAMETERS DETAILS

In our work, the feature layer $l$ is set to 3, and the momentum coefficient starts from $\xi = 0.999$ and is increased to 1 in a cosine curve. Also, AdAutoMix uses the same set of hyperparameters in all experiments as follows: $\alpha$=0.5, $\beta$=0.3, $\lambda$=1.0, $N$=3 or $N$=2.

## A.3 EXPERIMENTS IMPLEMENTATION DETAILS

On CIFAR100, RandomFlip and RandomCrop with 4-pixel padding are used as basic data augmentations for images with size $32 \times 32$. For ResNet18 and ResNeXt50, we use the following experimental setting: SGD optimizer with momentum of 0.9, weight decay of 0.0001, batch size of 100, and training with 800 epochs. The basic learning rate is 0.1, dynamically adjusted by the cosine scheduler; CIFAR version of ResNet variants are used, *i.e.*, replacing the $7 \times 7$ convolution and MaxPooling by a $3 \times 3$ convolution. To train Vit-based approaches, *e.g.* Swin-Tiny Transformer, we resize images to $224 \times 224$ and train them with AdamW optimizer with weight decay of 0.05, batch size of 100, and total training 200 epochs. The basic learning rate is 0.0005, dynamically adjusted by the cosine scheduler. For ConvNeXt-Tiny training, the images keep the $32 \times 32$ resolution, and we train it based on the setting of Vit-based approaches except for the basic learning rate of 0.002. the $\alpha$ and $\beta$ are set to 0.5 and 0.3 for CIFAR on ResNet18 and ResNeXt50.

On Tiny-ImageNet, RandomFlip and RandomResizedCrop for $64 \times 64$ are used as basic data augmenting. Except for a learning rate of 0.2 and training over 400 epochs, training settings are similar to the ones used in CIFAR100.

On ImageNet-1K, we use a Pytorch-style training setup, which optimizes the model for 100 epochs by SGD optimizer with a batch size of 256, a basic learning rate of 0.1, the SGD weight decay of 0.0001, and the SGD momentum of 0.9.

On CUB-200, FVGC-Aircrafts and Standford-Cars, we use the official PyTorch pre-trained models on ImageNet-1k are adopted as initialization, using SGD optimizer with momentum of 0.9, weight decay of 0.0005, batch size of 16, 200 epochs, learning rate of 0.001, dynamically adjusted by cosine scheduler. the $\alpha$ and $\beta$ are set to 0.5 and 0.1.

## A.4 DETAILS OF THE EXPERIMENTS FOR THE OTHER MIXUP

You can access detailed experimental settings and results at https://github.com/Westlake-AI/openmixup, which also provides the open-source code for most of the compared Mixup methods.

## A.5 RESULTS OF CALIBRATION

Table 6: The expected calibration error (ECE) of ResNet18 and Swin-Tiny Transformer (Swin-Tiny) with various Mixup methods trained on CIFAR100 dataset for 200 epochs.

| Classifiers | Mixup | CutMix | FMix | GridMix | PuzzleMix | AutoMix | AdAutoMix |
|---|---|---|---|---|---|---|---|
| ResNet18 | 15.3 | 4.4 | 8.9 | 6.5 | 3.7 | 3.4 | **3.2** (-0.2) |
| Swin-Tiny | 13.4 | 10.1 | **9.2** | 9.3 | 16.7 | 10.5 | **9.2** (-0.0) |

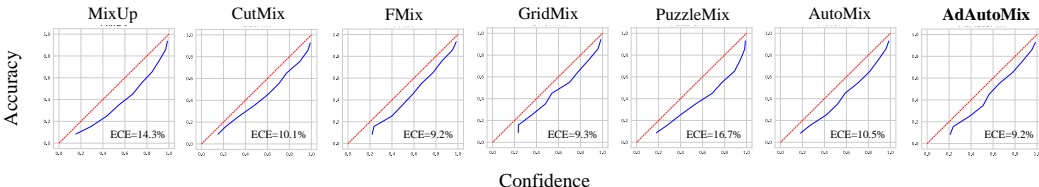

Figure 7: *Calibration* plots of Mixup variants and AdAutoMix on CIFAR-100 using ResNet-18. The red line indicates the expected prediction tendency.

## A.6 THE ACCURACY OF VARIOUS MIXUP APPROACHES ON OCCLUSION IMAGE SET

Table 7: The accuracies of ResNet50 and Swin-Tiny Transformer trained by various Mixup approaches on CIFAR100 and CUB200 datasets with different occlusion ratios.

| Method | 0% | 10% | 20% | 30% | 40% | 50% | 60% | 70% | 80% | 90% |
|---|---|---|---|---|---|---|---|---|---|---|
| | Swin-Tiny Transformer on CIFAR100 | | | | | | | | | |
| MixUP | 76.82 | 74.54 | 71.88 | 67.98 | 63.18 | 55.26 | 44.20 | 30.07 | 15.69 | 6.14 |
| PuzzleMix | 80.45 | 78.98 | 77.52 | 75.47 | 71.16 | 64.42 | 53.40 | 38.53 | 21.39 | 7.91 |
| AutoMix | 82.68 | 81.40 | 79.05 | 75.44 | 70.61 | 64.30 | 55.25 | 40.92 | 23.09 | 9.73 |
| AdAutoMix | 84.33 | 82.41 | 80.16 | 76.84 | 72.09 | 66.74 | 58.09 | 46.48 | 28.02 | 9.91 |
| | ResNet-50 on CUB200 | | | | | | | | | |
| Method | 0% | 10% | 20% | 30% | 40% | 50% | 60% | 70% | 80% | 90% |
| Vanilla | 82.15 | 74.75 | 61.89 | 46.24 | 30.81 | 16.67 | 8.94 | 4.63 | 2.23 | 1.07 |
| CutMix | 83.05 | 76.45 | 64.44 | 50.86 | 39.47 | 28.99 | 20.78 | 14.46 | 8.64 | 2.21 |
| PuzzleMix | 84.01 | 80.99 | 76.01 | 68.45 | 58.15 | 43.44 | 28.41 | 15.38 | 5.76 | 2.39 |
| AutoMix | 84.10 | 81.90 | 78.05 | 73.18 | 64.96 | 51.21 | 36.85 | 22.35 | 8.63 | 3.88 |
| AdAutoMix | 84.57 | 82.46 | 80.16 | 75.84 | 66.19 | 55.74 | 40.19 | 25.44 | 10.04 | 4.39 |

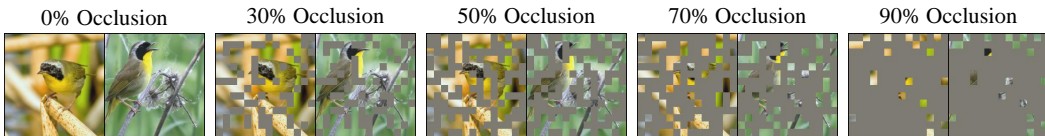

Figure 8: The images with different occlusion ratios.

## A.7 THE CURVES OF EFFICIENCY AGAINST ACCURACY

The training time of various mixup data augmentation approaches against accuracy is shown in Fig. 9. AdAutoMix take more computation time, but it consistently outperforms previous state-of-the-art methods with different ResNet architectures on different datasets.

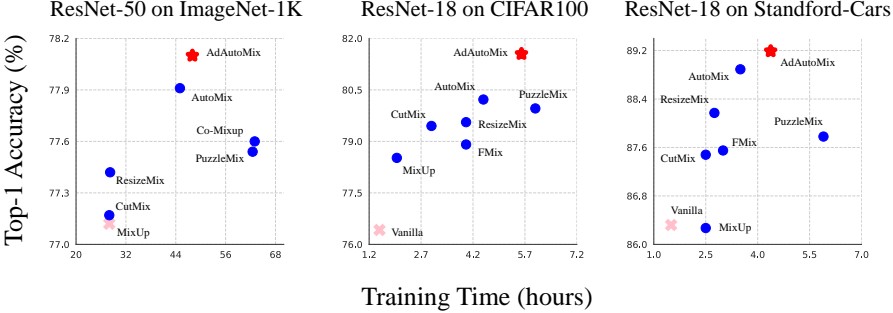

Figure 9: The plot of efficiency vs. accuracy

## A.8 ADAUTOMIX MODULES EXPERIMENT

Table 8 lists the accuracy of our AdAutoMix by gradually increasing regularization terms. The experimental results imply that each regularization term is capable of improving the robustness of AdAutoMix.

Table 9 shows the accuracy of our AdAutoMix with different $\lambda$. The experimental results show that AdAutoMix with $\lambda = 1$ as default achieves the best performances on CIFAR100 dataset.

Table 10 shows the accuracy of our AdAutoMix with different input image number $N$. From Table 10, we can see that AdAutoMix achieve the highest accuracy with $N=3$ on CIFAR100.

Table 8: Loss function experiments on CIFAR100 based on ResNet18.

| Method | Base | Base+0.5L$_{ace}$ | Base+0.5L$_{ace}$ + 0.5L$_{mce}$ | Base+0.5L$_{ace}$ + 0.5L$_{mce}$ $-0.3$L$_{amce}$ | Base+0.5L$_{ace}$ + 0.5L$_{mce}$ $-0.3$L$_{amce}$ + 0.7L$_{cosine}$ |
|---|---|---|---|---|---|
| ResNet18 | 79.38 | 79.98 | 80.04 | 81.32 | 81.55 |

Table 9: Classification accuracy of ResNet 18 with different $\lambda$ ratio.

| | CIFAR100 | | | | |
|---|---|---|---|---|---|
| Method | 0.2 | 1.0 | 2.0 | 5.0 | 10.0 |
| ResNet18 | 82.27 | **82.32** | 81.73 | 80.02 | 81.05 |
| ResNeXt50 | 84.22 | **84.40** | 83.99 | 84.31 | 83.63 |

Table 10: Classification accuracy of ResNet18 trained by AdAutoMix with different input image number $N$, where $N = 1$ means that it is vanilla method.

| | CIFAR100 | | |
|---|---|---|---|
| inputs | Top1-Acc(%) | Top5-Acc(%) | Times s/iter |
| N = 1 | 78.04 | 94.60 | 0.1584 |
| N = 2 | 82.16 | 95.88 | 0.1796 |
| N = 3 | **82.32** | **95.92** | **0.2418** |
| N = 4 | 81.78 | 95.68 | 0.2608 |
| N = 5 | 80.79 | 95.80 | 0.2786 |

### A.9 ACCURACY OF RESNET-18 TRAINED BY ADAUTOMIX WITH AND WITHOUT ADVERSARIAL METHODS.

Figure 10 shows the accuracy of ResNet-18 trained by our AdAutoMix with and without adversarial training on CIFAR100. The experimental results demonstrate AdAutoMix with adversarial training achieves higher classification accuracy on CIFAR100 dataset, which implies that the proposed adversarial framework is capable of generating harder samples to improve the robustness of classifier.

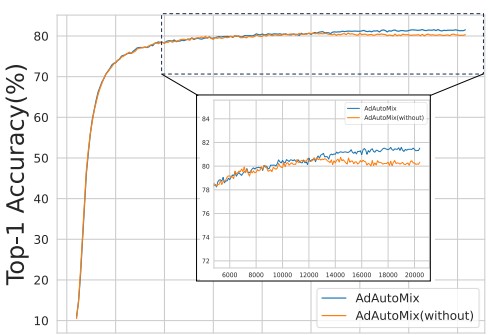

Figure 10: The Top-1 accuracy plot of AdAutoMix training with and without adversarial methods.

### A.10 COMPARISON WITH OTHER ADVERSARIAL DATA AUGMENTATION

We further compare Mixup (Zhang et al., 2017) and our AdAutoMix with existing Adversarial Data Augmentation methods, *e.g.* DADA (Li et al., 2020), ME-ADA (Zhao et al., 2020), and SAMix (Zhang et al., 2023). Table 11 depicts the classification accuracy of various approaches. The experimental results in Table 11 demonstrate that our AdAutoMix outperforms existing Adversarial Data Augmentation methods and achieves the highest accuracy on the CIFAR100 dataset.

Table 11: Experiments with AdAutoMix and other Adversarial Data Augmentation methods.

|  | Baseline | MixUp | DADA | ME-ADA | SAMix | AdAutoMix |
|---|---|---|---|---|---|---|
| ResNet-18 | 76.42 | 78.52 | 78.86 | 77.45 | 54.01 | **81.55** |

### A.11 Algorithm of AdAutoMix

---

**Algorithm 1** AdAutoMix training process

---

**Input:** Encoder $E_\phi, E_{\widehat{\phi}}$, Classifier $\psi_W, \psi_{\widehat{W}}$, Samples $\mathbb{S}$, lambda $\lambda$, Generator $G_\theta(\cdot)$, coefficient $\xi$
and feature map $z_n^l$
1: $E_{\widehat{\phi}}.parmas = E_\phi.params$
2: **for** $\mathbb{X}, \mathbb{Y}$ in $\mathbb{S}$ loder **do**
3:      $z_n^l = E_{\widehat{\phi}}(\mathbb{X})$
4:      $x_{mix} = G_\theta(z_n^l, \lambda)$
5:      $L_{amce} = \psi_W(x_{mix}, \lambda, \mathbb{Y})$
6:      $L_{\widehat{amce}}, L_{cosine} = \psi_{\widehat{W}}(x_{mix}, \lambda, \mathbb{Y})$
7:      **for** $1 < t_1 < T_1$ **do**
8:          $Update\ W(t+1)$ according to Eq.14
9:      **end for**
10:      **for** $1 < t_2 < T_2$ **do**
11:          $Update\ \theta(t+1)$ according to Eq.16
12:      **end for**
13:      $Update(E_{\widehat{\phi}}.params, E_\phi.params)$
14:      $E_{\widehat{\phi}}.params = \xi * E_{\widehat{\phi}}.params + (1 - \xi) * E_\phi.params$
15: **end for**

---

## B Visualization of Mixed Samples

### B.1 Class activation mapping (CAM) of different mixup samples.

The Class activation mapping (CAM) of various Mixup models are shown in Fig. 11.

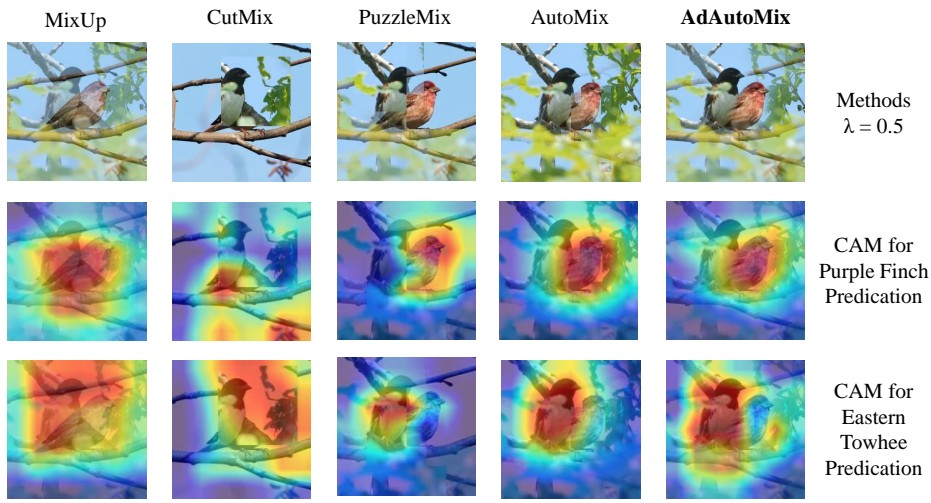

Figure 11: The class activation map of various Mixup models ($\lambda = 0.5$).

## B.2 MIXED SAMPLES ON CUB-200

The mixed samples generated by our approach trained on CUB200 dataset are depicted in Fig. 12.

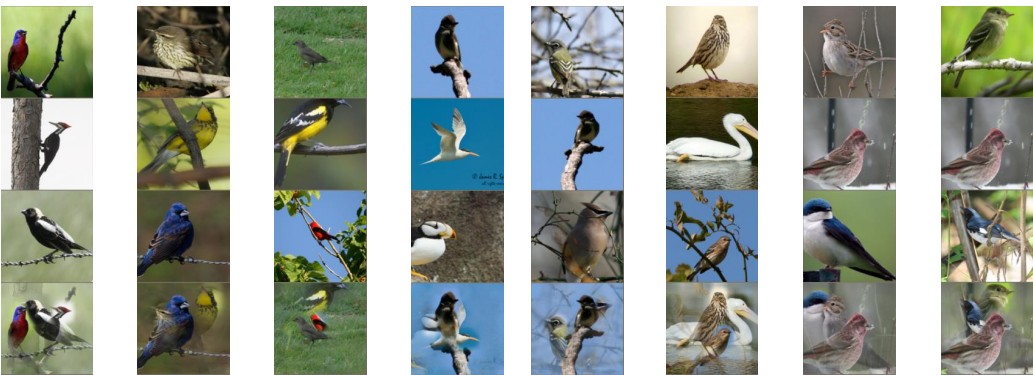

Figure 12: Visualization of mixed samples on CUB-200.

## B.3 MIXED SAMPLES ON CIFAR100

The mixed samples generated by our approach trained on CIFAR100 dataset are shown in Fig. 13.

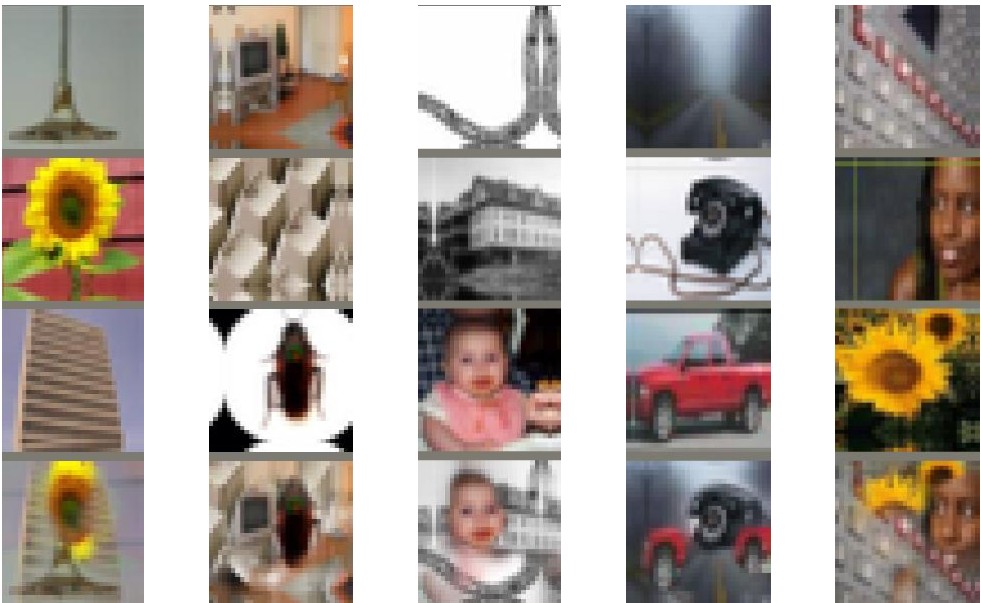

Figure 13: Visualization of mixed samples on CIFAR100.

## B.4  DIVERSITY OF SAMPLES GENERATED BY VARIOUS APPROACHES

To demonstrate that AdAutoMix is capable of generating diversity samples, we show the synthesising images of AdAutoMix and AutoMix on ImageNet-1K. From Fig. 14, we can see that AdAutoMix produces mixed samples with more differences. By contrast, AutoMix generates similar images at different iteration epochs, which implies that the proposed AdAutoMix has the capacity to produce diverse images by adversarial training.

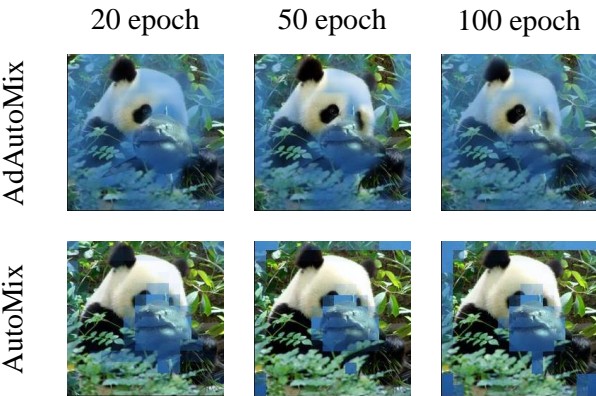

Figure 14: Mixed samples with different epochs.

