# OpenReview forum: "Adversarial AutoMixup"
_ICLR.cc/2024/Conference — ICLR 2024 spotlight_

### Official Review · Reviewer_sHE4 · 2023-10-29

**Soundness:** 4 excellent
**Presentation:** 3 good
**Contribution:** 4 excellent
**Rating:** 8
**Confidence:** 4

**Summary:**

The paper proposes a new augmentation technique. First it proposes hard samples to train and secondly a robustification of the classifier. The method is evaluated on 4 datasets

**Strengths:**

The idea to augment with hard examples is interesting. Furthermore, to iterate between augmentation and classifier is also interesting.
Showed results are strong.

**Weaknesses:**

I do not see any significant weakness. The method is harder to implement and it requires more resources that other augmentation techniques, but given the timeline of augmentation, it is expected

**Questions:**

None. I see this paper as a clear contribution.

---

> ### Author Response · Authors · 2023-11-18
>
> **A1:Thank you for your comments on our work.**
>
> In line with the point you raise, we have added a plot of efficiency against accuracy in Figure 9 in the Appendix in page 16.  From Figure 9, we observe that our approach takes more time than AutoMix for training but it achieves the best performance w.r.t the existing approaches.  For testing, nonetheless, there is no difference between AutoMix and AdAutoMix in terms of time cost.

---

### Official Review · Reviewer_xmzW · 2023-10-30

**Soundness:** 3 good
**Presentation:** 3 good
**Contribution:** 2 fair
**Rating:** 6
**Confidence:** 3

**Summary:**

The paper proposes an adversarial data augmentation strategy that builds on top of AutoMix. The framework alternates between training a mixed example generator and a target classifier. The mixed sample generator aims to produce hard examples and the target tries to improve generalization by learning robust features. With automatic mixup approaches (based on saliency or otherwise) the combination is deterministically selected, and there is no sample diversification. To mitigate this, the method proposes an adversarial generator instead.
To prevent a collapse from the generator, an EMA teacher and a weighted cosine similarity term between the mixed sample and individual samples is used for end-to-end learning.

**Strengths:**

Results are consistently better than AutoMixup and the evaluation (Table 1) is thorough.



--------
Post-rebuttal:

The authors have adequately addressed the concerns in the review. Useful experiments and ablations have been added as well. I'm still a little skeptical about the actual impact of the paper, from the methods and corresponding evaluation numbers in the paper I believe that we're at the point of diminishing returns.

I've therefore increased my score to a 6.

**Weaknesses:**

There is no evaluation compared to Adversarial data augmentation approaches [1, 2, 3, 4]. At least an introduction or related works section should be added as relevant approaches to the problem.

The term “cross attention module” (CAM) should not be used as it can be confused with “class activation mapping” (CAM) which is generally used in saliency-based data augmentation methods.

Some notation is confusing - the encoder weight is updated with an EMA of the weights of the classifier - $\hat{\phi} = \xi \hat{\phi} + (1-\xi) W$. Is it unclear if the encoder refers to the generator or the classifier. Later near Equation 12, $\psi$ is referred to as a target classifier with weights $W$.

Equation 7 and 8 refer to the same value of $y$ used in cross entropy. It is better to keep the form of the loss consistent, since $y_{mix} = \sum_i y_i \lambda_i$, implies

$\sum_i L_{ce}(\psi(x_{mix}), y_i) \lambda_i = \sum_i -\lambda_i y_i \log(\psi(x_{mix})) = \log(\psi(x_{mix})) \sum_i -\lambda_i y_i = -\log(\psi(x_{mix})) y_{mix} = L_{ce}((\psi(x_{mix}), y_{mix}))$


Equations 10 through 15 are badly formatted and hard to read. It is also unclear what the individual contribution of the four cross-entropy terms are, and a suitable way to choose $\alpha$ and $\beta$.

Section 4.2 mentions the proposed method has the lowest calibration error, but there is no table showing the ECE of other baselines. Fig.4. shows the ECE of only the proposed method.


Typos and minor mistakes:
“to facility representation”
Mxiup → mixup
bad formatting  in eq 5
notation in eq 6, 7 is unclear. is the $*$ scalar multiplication?
what is meant by “inherent meaning of images” - this sounds slightly unscientific, this should be explained in a bit more detail

Currently, I think the paper needs a lot of work - both in terms of coherence and motivation for the method. There are too many elements all over the place and it is unclear what the improvement actually comes from. The evaluation criteria is not standard (see Questions) and needs more justification. Therefore, I recommend an initial reject.

_______


[1] Zhang, Jiajin, et al. "Spectral Adversarial MixUp for Few-Shot Unsupervised Domain Adaptation." International Conference on Medical Image Computing and Computer-Assisted Intervention. Cham: Springer Nature Switzerland, 2023.

[2] Xu, Minghao, et al. "Adversarial domain adaptation with domain mixup." Proceedings of the AAAI conference on artificial intelligence. Vol. 34. No. 04. 2020.

[3] Zhao, Long, et al. "Maximum-entropy adversarial data augmentation for improved generalization and robustness." Advances in Neural Information Processing Systems 33 (2020): 14435-14447.

[4] Antoniou, Antreas, Amos Storkey, and Harrison Edwards. "Data augmentation generative adversarial networks." arXiv preprint arXiv:1711.04340 (2017).

**Questions:**

The method compares the median of the top-1 test accuracy in the last 10 training epochs. Since adversarial methods are generally brittle and have unstable training dynamics, does the mean test accuracy fluctuate a lot? Also, it seems that there is no validation set used to choose the best checkpoint. This evaluation criteria is not justified in the paper.

---

> ### Author Response · Authors · 2023-11-18
>
> **W1: There is no evaluation compared to Adversarial data augmentation approaches [1, 2, 3, 4]. At least an introduction or related works section should be added as relevant approaches to the problem.**
>
> **A1:** Thanks for your suggestions. We have included the performance of adversarial data augmentation approaches [1, 2, 3, 4] in comparison experiments. Also, we added them in the related works’ section in subsection 2.3 in page 3.
> In work[1], a Sensitivity-guided Spectral Adversarial MixUp (SAMix) method is proposed to generate target-style images for domain adaptation. Similarly, work[2]  proposes an adversarial learning framework which maps both domains to a common latent distribution by domain mixup, and efficiently transfer knowledge learned on the supervised domain to its unsupervised counterpart. Work[3] investigates adversarial data argumentation from an information theory perspective and proposes a maximum-entropy regularization in the maximization phase of adversarial data augmentation. In work[4], a novel Generative Adversarial Network(GAN) is investigated to learn a representation and generate samples for data augmentation.
> Tu summarize, the adversarial mixup models in works[1][2] are proposed to generate mix features or images for unsupervised domain adaptation instead of classification. Work[3] investigates adversarial framework with maximum-entropy regularization to generate samples for classification.  Work[4] proposed a GAN for image generation and classification.  Therefore, we compared these works[3][4] with our model in terms of accuracy improvement. The classification accuracies are listed in Table 11 in Appendix in page 17. The results show that our approach outperforms the existing adversarial data augmentation approaches mentioned in the response.
>
> ***
> **W2:The term “cross attention module” (CAM) should not be used as it can be confused with “class activation mapping” (CAM) which is generally used in saliency-based data augmentation methods.**
>
> **A2:** Thanks for pointing this out. We have replaced the “cross attention module (CAM)” by “cross attention block (CAB)” in revised version.
>
> ***
> **W3: Some notation is confusing - the encoder weight is updated with an EMA of the weights of the classifier $\widehat\phi=\xi\widehat\phi+(1-\xi)W'$. Is it unclear if the encoder refers to the generator or the classifier. Later near Equation 12, $\psi$ is referred to as a target classifier with weights $W$.**
>
> **A3:** As shown in Figure 2, the generator consists of encoder $E_\phi$ and Mixed module (Figure 3). Note that the weights of the encoder are updated by an exponential moving average (EMA) of the target classifier $\widehat\phi=\xi\widehat\phi+(1-\xi)W'$, where $W'$ is the partial weight of target classifier. In our experiments, existing classifiers such as ResNet18, ResNet34, and ResNeXt50 are employed as target classifiers, with the weights of the first three layers employed to update encoder $E_\phi$ by EMA.  In this case, $W'$ is the weight of the first three layers in target classifier.  For target classifier  $\psi_W$, $W$ is referred to the weight of all layers in the target classifier. For clarification, we have added the above contents in subsection 3.2 at page 4.
>
> ***
> **W4: Equation 7 and 8 refer to the same value of y used in cross entropy. It is better to keep the form of the loss consistent.**
>
> **A4:** Thanks for your suggestions. We have modified Eq.3 to keep the form of the loss consistent in page 4.
> ***
> **Reference**
> [1] Zhang, Jiajin, et al. "Spectral Adversarial MixUp for Few-Shot Unsupervised Domain Adaptation." International Conference on Medical Image Computing and Computer-Assisted Intervention. Cham: Springer Nature Switzerland, 2023.
>
> [2] Xu, Minghao, et al. "Adversarial domain adaptation with domain mixup." Proceedings of the AAAI conference on artificial intelligence. Vol. 34. No. 04. 2020.
>
> [3] Zhao, Long, et al. "Maximum-entropy adversarial data augmentation for improved generalization and robustness." Advances in Neural Information Processing Systems 33 (2020): 14435-14447.
>
> [4] Antoniou, Antreas, Amos Storkey, and Harrison Edwards. "Data augmentation generative adversarial networks." arXiv preprint arXiv:1711.04340 (2017).

---

> > ### Author Response · Authors · 2023-11-18
> >
> > ***
> > **W5: Equations 10 through 15 are badly formatted and hard to read. It is also unclear what the individual contribution of the four cross-entropy terms are, and a suitable way to choose $\alpha$ and $\beta$.**
> >
> > **A5:** According to your suggestions, we have reformatted Equations 10 -15 to clarify the presentation. In addition, we have included the following contents in subsection 3.3.1 in page 6 to explain the individual contributions.
> >    “Notice that $L_{ce}(\psi_W, y)$  is the standard cross-entropy loss. $L_{ace}(\psi_W, \mathbb{Y})$ loss facilitates the backbone to provide a stable feature map at the early stage so that it speeds up convergence. Target loss $L_{amce}(\psi_W, \mathbb{Y})$  aims to learn task-relevant information in the generated mixed samples. $L_{mce}(\psi_W, y_{mix})$ facilitates the capture of task-relevant information in the original mixed samples.”
> >
> > Also, we have conducted ablation experiments by gradually adding the regularization losses and computed their accuracy. The experimental results are listed in Table 5 in page 9 and Table 8 in Appendix at page 16.
> >
> > Regarding the optimization of hyperparameters, existing mixup augmentation approaches [1-10] usually conduct comparable experiments on CIFAR-100 , Tiny-ImageNet, ImageNet-1K, CUB-200-2011,  FGVC-Aircrafts,  and  Standford-Cars,  as detailed in A.1 in the Appendix. However, these original datasets include only two subsets: either training set and  validation set or training set and test set. In these works [1-8], they treat the validation set as test set. So, they do not employ a validation set in the experiments and all models are trained on the training set and stopped with predefined epochs. Then, the accuracy on the test set or validation set are reported for comparison. Therefore, for fair comparison, we follow their experimental settings and do not use the validation set to choose the best checkpoint. Instead, similar to works SuperMix and AutoMix,  we provide the sensitivity of the hyperparameters N, λ, α, and β with the dataset (as shown in Figure 6) in the ablation experiments in page 9 .  More experimental results are included in Tables 9 and 10 in the Appendix at page 16.
> > We do think, nonetheless, that your suggestions are very valuable . In the future, we will split, for each dataset, the current  training set, into an actual training set and a validation set, while keeping the test set unchanged. The hyperparameters will be optimized on the validation set. Alternatively, we will optimize the hyperparameters by cross-validation on the training dataset.
> >
> > ***
> > **W6: Section 4.2 mentions the proposed method has the lowest calibration error, but there is no table showing the ECE of other baselines. Fig.4. shows the ECE of only the proposed method.**
> >
> > **A6:** In the previous version, the ECE of the tested approaches were shown in the corresponding sub-figs in Figure 4. According to your suggestion, we show, in the revised vision,  the ECE in Table 6 in Appendix in page 14 and Figure 4 in page 8.
> > |Classifiers|MixUp|CutMix|FMix|GridMix|PuzzleMix|AutoMix|AdAutoMix|
> > |---|:--:|:--:|:--:|:--:|:--:|:--:|:--:|
> > |ResNet-18|15.3|4.4|8.9|6.5|3.7|3.4|3.2(-0.2)|
> > |Swin-Tiny Transformer|13.4|10.1|9.2|9.3|16.7|10.5|9.2(-0.0)|
> >
> > ***
> > **W7: Typos and minor mistakes: “to facility representation” Mxiup → mixup bad formatting in eq 5 notation in eq 6, 7 is unclear. is the ∗ scalar multiplication? what is meant by “inherent meaning of images” - this sounds slightly unscientific, this should be explained in a bit more detail.**
> >
> > **A7:** Thanks for pointing this out. We have corrected the “to facility representation” to “to facilitate representation”. The “Mxiup ” is changed to “mixup”. Eq.5 have been reformatted. Also, we have provided the explanation  that the notation ∗  denotes scalar multiplication in page 4 in the revised version.
> > In our work, the inherent meaning of images is related to the the semantic meaning of the images. For example,  if there is cat in an image,  the label of cat is the semantic meaning.  If we fully mask the regions associated with cat, there is no pattern for classification. We have added the explanation in the following sentence at page 6.
> > “It is possible, therefore, that the inherent meaning of images (i.e.  the semantic meaning of the images) collapse”

---

> > > ### Author Response · Authors · 2023-11-18
> > >
> > > **W8: Currently, I think the paper needs a lot of work - both in terms of coherence and motivation for the method. There are too many elements all over the place and it is unclear what the improvement actually comes from. The evaluation criteria is not standard (see Questions) and needs more justification. Therefore, I recommend an initial reject.**
> > >
> > > **A8:** Your concerns give us the opportunity to clarify this out. We have tried our best to revise the paper so as to improve the coherence and motivation of our approach.
> > > There are several contributions for our work:
> > > 1) The major contribution is that we design a novel adversarial framework to generate hard samples, so as to avoid overfitting.  We have shown the accuracy of AdAutomixup with and without adversarial training in Figure 10 in page 16 and the experimental results show that our AdAutomixup model train a more robust classifier for image classification and outperforms current state of the art.
> > > 2) The second contribution is that our AdAutomixup extends the images’ mixing from  two images to multiple ones.  In our ablation experiments, we have computed the accuracy of our model with different numbers of mixed images, with the results listed in Figure 6(c) in page 9 and Table 10 in the Appendix in page 16. The experimental results show that the best performance is achieved by mixing three images instead of two images.
> > > 3) The third contribution is that we introduce regularization terms, i.e. $L_{mce}(\psi_W, y_(mix))$, $L_{ace}(\psi_W, \mathbb{Y})$, $L_{amce}(\psi_\widehat{W}, \mathbb{Y})$  and $L_{cosine}$ to improve the robustness of our model.  To investigate the impact of each one, we show the accuracy of our model by adding gradually the regularization terms in ablation experiments. The experimental results are listed in Table 5 in page 9.  For example, the two modules $L_{mce}(\psi_W,y_{mix})$ and $L_{ace}(\psi_W,\mathbb{Y})$ increase classifier accuracy by about 0.66%.  Further, the performance is improved by adding the two blocks $L_{amce}(\psi_\widehat{W},\mathbb{Y})$ and $L_{cosine}$. Also, we have shown the improvement of each element in Table 8 in the Appendix.  These experimental results demonstrate that each regularization term improves the robustness of our AdAutoMix model.
> > > Regarding the evaluation criteria, we have followed the experimental settings of AutoMix [1] for fair comparison.  For example, the top-1 test accuracy in the last 10 training epochs are reported, and all models proposed by the other works are trained with the same number of epochs.  In fact, many existing works [1-10] in this area employed top-1 test accuracy as  evaluation criterion.
> > >
> > > ***
> > > **reference**
> > > [1] Liu, Zicheng et al. “AutoMix: Unveiling the Power of Mixup for Stronger Classifiers.” European Conference on Computer Vision (2021).
> > > [2] Liu, Zicheng, et al. "Harnessing hard mixed samples with decoupled regularizer." Thirty-seventh Conference on Neural Information Processing Systems. 2023.
> > > [3] Kim, Jang-Hyun, Wonho Choo, and Hyun Oh Song. "Puzzle mix: Exploiting saliency and local statistics for optimal mixup." International Conference on Machine Learning. PMLR, 2020.
> > > [4] Kim, JangHyun, et al. "Co-Mixup: Saliency Guided Joint Mixup with Supermodular Diversity." International Conference on Learning Representations. 2020.
> > > [5] Chen, Mengzhao, et al. "Smmix: Self-motivated image mixing for vision transformers." Proceedings of the IEEE/CVF International Conference on Computer Vision. 2023.
> > > [6] Venkataramanan, Shashanka, et al. "Alignmixup: Improving representations by interpolating aligned features." Proceedings of the IEEE/CVF Conference on Computer Vision and Pattern Recognition. 2022.
> > > [7] Li, Siyuan, et al. "Boosting discriminative visual representation learning with scenario-agnostic mixup." arXiv preprint arXiv:2111.15454 (2021).
> > > [8] Zhao, Qihao, et al. "MixPro: Data Augmentation with MaskMix and Progressive Attention Labeling for Vision Transformer." The Eleventh International Conference on Learning Representations. 2022.
> > > [9] Chen, Jie-Neng, et al. "Transmix: Attend to mix for vision transformers." Proceedings of the IEEE/CVF Conference on Computer Vision and Pattern Recognition. 2022.
> > > [10] Dabouei, Ali, et al. "Supermix: Supervising the mixing data augmentation." Proceedings of the IEEE/CVF conference on computer vision and pattern recognition. 2021.

---

> > > > ### Author Response · Authors · 2023-11-18
> > > >
> > > > **Q1: The method compares the median of the top-1 test accuracy in the last 10 training epochs. Since adversarial methods are generally brittle and have unstable training dynamics, does the mean test accuracy fluctuate a lot? Also, it seems that there is no validation set used to choose the best checkpoint. This evaluation criteria is not justified in the paper.**
> > > >
> > > > **Answer:** Thanks for your questions.  In fact, we compare the mean performance of 3 trials where the median of top-1 test accuracy in the last 10 training epochs is recorded for each trial. In existing adversarial methods, various strategy are employed to train a stable model. In our approach, an exponential moving average (EMA) teacher and a cosine similarity are employed as regularization to reduce the search space, so as to accelerate convergence. As shown in Figure 1 and Figure 10 in the Appendix, the curve for our approach which plotted the accuracy against the iteration step is smooth and stable after 7500 iterations, so there are less fluctuations for the mean performance of 3 trials.
> > > >
> > > > It is important to note that, to achieve fair comparison, we follow the same experimental settings of state of the art method, AutoMix [1].   For example, our approach is trained for the same number of epochs in comparative experiments and the mean performances of 3 trials are reported for comparison.  In fact, existing Mixup augmentation approaches, such as PuzzleMix[3], TransMix[9], and SuperMix[10], employ same experimental setting.
> > > >
> > > > Regarding the optimization of hyperparameters, most existing mixup augmentation approaches [1-10] usually conduct comparable experiments on CIFAR-100 dataset, Tiny-ImageNet, ImageNet-1K,CUB-200-2011,  FGVC-Aircrafts,  and  Standford-Cars,  as detailed in A.1 in the Appendix. However, these original datasets include only two subsets: training set and validation set or training test and test set. In these works, they treat the validation set as test set. So, they do not employ a validation set for hyperparameters’ optimization and all models are trained on the training set and stopped with a predefined number of epochs. Then, the accuracy on the test set (or validation set, treated as test set) is reported for comparison. Therefore, for fair comparison, we follow their experimental settings and do not use the validation set to choose the best checkpoint. Instead, similar to works SuperMix and AutoMix,  we provide the sensitivity of the hyperparameters N, λ, α, and β with the dataset (as shown in Figure 6) in the ablation experiments in page 9.   More experimental results are included in Tables 9 and 10 in the Appendix in page 16.
> > > >
> > > > We do think, nonetheless, that your suggestions are very valuable . In the future, we will split, for each dataset, the current  training set, into an actual training set and a validation set, while keeping the test set unchanged. The hyperparameters will be optimized on the validation set. Alternatively, we will optimize the hyperparameters by cross-validation on the training dataset.

---

> > > > > ### Author Response · Authors · 2023-11-20
> > > > >
> > > > > **Dear Reviewer,**
> > > > > **We sincerely appreciate your dedicated effort in reviewing our manuscript. However, unlike previous years, there is no secondary author-reviewer discussion this year, and you have until November 22, 2023, to provide your recommendations.
> > > > > Given the proximity of this date, we eagerly anticipate hearing from you. We have thoroughly considered all the feedback and provided a detailed response a few days ago. We hope our response sufficiently addresses your questions. Please don't hesitate to let us know if you still require clarification or have additional questions.**
> > > > >
> > > > > **Warm regards,**
> > > > > **Authors**

---

> > > > > > ### Comment · Reviewer_xmzW · 2023-11-20
> > > > > > **Feedback to Rebuttal**
> > > > > >
> > > > > > Dear authors,
> > > > > >
> > > > > > Thank you for the (very) detailed feedback. Apologies for the late response as I'm travelling. I had a look at the revised paper and other reviews and feedback, and have decided to increase my score to weak accept.
> > > > > >
> > > > > > The paper is highly reformatted as per the reviews, and reads much crisper. Experiment setup is as thorough as it can be.
> > > > > >
> > > > > > Small Questions/Clarifications:
> > > > > >
> > > > > > 1. The proposed module is motivated by the generation of robust samples using adversarial training, which AutoMix may not achieve. Is it feasible to show samples generated by the proposed method (over iterations), whose latents are farthest from the training set, and see how diverse these examples are. AutoMix would probably show examples that are very similar mixes.

---

> > > > > > > ### Author Response · Authors · 2023-11-23
> > > > > > >
> > > > > > > Dear Reviewer,
> > > > > > > **Thank you very much for your recognition of our work. The problems you mentioned before were very helpful to us, and we have taken them very seriously and have improved them as suggested.**
> > > > > > > For the problem you mentioned now, **"Is it feasible to show samples generated by the proposed method (over iterations), whose latents are farthest from the training set, and see how diverse these examples are".**
> > > > > > >  We were pleasantly surprised by this issue and think it’s a good one worth addressing. We have placed mixed samples from AdAutoMix and AutoMix at different times **on page 19**. As can be seen from the figure, in the later stages of training, **AutoMix’s changes to the samples are limited to just around the features, whereas AdAutoMix’s changes in the later stages of training are more pronounced as compared to the early stages of training due to adversarial means, and are arguably more “chaotic” and challenging.**
> > > > > > > **In future work, we will take this issue more into account and make more detailed experiments.**
> > > > > > > Finally, thank you again for recognising our work! We also hope that our work will bring some inspiration and help to other researchers.
> > > > > > >
> > > > > > > Best regards,
> > > > > > > Authors.

---

### Official Review · Reviewer_vYYM · 2023-11-06

**Soundness:** 3 good
**Presentation:** 3 good
**Contribution:** 3 good
**Rating:** 8
**Confidence:** 5

**Summary:**

Mixup data augmentations are widely used and usually require well-designed sample mixing strategies, e.g., AutoMix optimized in an end-to-end manner. However, using the same mixup classification loss as the learning objective for both the mixed sample generation and classification tasks might cause consistent and unitary samples, which lack diversity. Based on AutoMix, this paper proposes AdAutomixup, an adversarial automatic mixup augmentation approach that generates challenging samples to train a robust vein classifier for palm-vein identification by alternatively optimizing the classifier and the mixup sample generator. Meanwhile, the authors introduce an EMA teacher with cosine similarity to train AdAutomixup preventing the collapse of the inherent meanings of images. Extensive experiments on five mixup classification benchmarks demonstrate the effectiveness of the proposed methods.

**Strengths:**

* (**S1**) This paper provides an interesting view of improving mixed sample qualities through adversarial training in the close-loop optimized mixup augmentation framework. The overall presentation of the manuscript is easy to follow, and the proposed methods are well-motivated.

* (**S2**) Extensive experiments on mixup benchmarks verify the performance gains of the proposed AdAutoMix compared to existing mixup methods. Popular Transformer architectures are included in experiments.

**Weaknesses:**

* (**W1**) More empirical analysis of the proposed methods can be added. Despite the authors visualizing the mixed samples and CAM maps of various mixup methods, it can only reflect the overall performances and characteristics of different methods. I suggest the authors provide a fine-grained analysis of each proposed module to demonstrate its effectiveness, e.g., plotting the classification accuracy of using adversarial training or not.

* (**W2**) Small-scale experiments. The authors only provide comparison results on CIFAR-10/100, Tiny-ImageNet, and fine-grained classification datasets. More experiments on ImageNet-1K or other large-scale datasets are required. Meanwhile, the evaluation tasks or metrics can be more diverse, such as more robustness evaluations with adversarial attacks and transfer experiments to downstream tasks.

* (**W3**) Some minor drawbacks in writing formats, and I suggest the authors take more time to polish the writing. As for Section 3, the arrangement of Sec. 3.1 and Sec. 3.2 can be reversed. Or the authors can provide a Preliminary section to introduce the background knowledge (e.g., mixup classification problem). As for equations, the text subscripts (e.g., $argmin_{\theta}$, $L_{amce}$) should be in bold format, i.e., using `\mathrm{}` as $\mathrm{argmin}_{\theta}$. As for tables and figures, there are some untidy arrangements, like Table 3, 4, and 5, and Figure 5 and 6. The author might zoom in on Figure 1 to show the detailed improvement of AdAutoMix.

================== Post-rebuttal Feedback ==================

Since the rebuttal feedback and revised version have almost addressed the weaknesses and concerns I mentioned, I raise my score to 8 and encourage the authors to further polish the writing issues and add more discussion of the limitations & future works.

**Questions:**

* (**Q1**) Do the authors provide the hyper-parameter settings of AdAutoMix (e.g., the mixing ratio $\lambda$, the mixed sample number $N$, and $\beta$ in Eq. (12)? The authors might provide a sensitivity analysis of the hyper-parameters in the Appendix.

---

> ### Author Response · Authors · 2023-11-18
>
> **W1: More empirical analysis of the proposed methods can be added. Despite the authors visualizing the mixed samples and CAM maps of various mixup methods, it can only reflect the overall performances and characteristics of different methods. I suggest the authors provide a fine-grained analysis of each proposed module to demonstrate its effectiveness, e.g., plotting the classification accuracy of using adversarial training or not.**
>
> **A1:** Thank your valuable suggestions. We have plotted the classification accuracy before and after using adversarial training. The classification accuracy is illustrated in Figure 10 in Appendix A.9 The experimental results imply that the proposed adversarial framework is capable of generating harder samples to improve the robustness of the classifier.
> |model|wo adversarial|w adversarial|
> |---|:--:|:--:|
> |ResNet-18|80.45|81.55|
> |ResNeXt-50|84.12|84.40|
> ***
> **W2: Small-scale experiments. The authors only provide comparison results on CIFAR-10/100, Tiny-ImageNet, and fine-grained classification datasets. More experiments on ImageNet-1K or other large-scale datasets are required. Meanwhile, the evaluation tasks or metrics can be more diverse, such as more robustness evaluations with adversarial attacks and transfer experiments to downstream tasks.**
>
> **A2:** According to your suggestions, we have carried out comparable experiments on the ImageNet-1K dataset to evaluate the performance of *AdAutoMix*, and the results are listed in Table 1 in page 7. Also, we have shown the robustness evaluations of our approach with adversarial attacks and transfer experiments. We have added a paragraph **“Occlusion Robustness”** in subsection 4.4 and the responding results are shown in Figure 5 in page 9 and Table 7 in Appendix A.6 in page 15. Also, we have done experiments to test our approach against **adversarial attacks** and the corresponding results are detailed in Table 4 in page 9. In addition, we have added **transfer learning** experiments in subsection 4.5, and the accuracy of various approaches are depicted in Table 3 in page 9.
>
> ***
> **W3: Some minor drawbacks in writing formats, and I suggest the authors take more time to polish the writing. As for Section 3, the arrangement of Sec. 3.1 and Sec. 3.2 can be reversed. Or the authors can provide a Preliminary section to introduce the background knowledge (e.g., mixup classification problem). As for equations, the text subscripts (e.g., argminθ, Lamce) should be in bold format, i.e., using \mathrm{} as argminθ. As for tables and figures, there are some untidy arrangements, like Table 3, 4, and 5, and Figure 5 and 6. The author might zoom in on Figure 1 to show the detailed improvement of AdAutoMix.**
>
> **A3:** Thanks for your suggestions to help us improve the quality of our work.
> - We have polished the English writing and carefully checked the grammar errors through the whole manuscript.
> - Because of limited space, we did not add a new Preliminary section but have reversed Sec. 3.1 and Sec. 3.2  in the original version (at page 4).
> - We have marked text subscripts in all equations in bold format. Also, we have rearranged Table 3,  4, and 5, and Figure 5 and 6 in the revised version in pages 8 and 9. In addition, Figure 1 is zoomed in to clarify the performance improvement of our *AdAutoMix* in page 2.
>
> ***
> **Q1:Do the authors provide the hyper-parameter settings of AdAutoMix (e.g., the mixing ratio $\lambda$, the mixed sample number N, and $\beta$ in Eq.(12)? The authors might provide a sensitivity analysis of the hyper-parameters in the Appendix.**
>
> **A4:** In the previous version, we have shown the hyper-parameter settings of AdAutoMix with the mixed sample number N, $\alpha$ and $\beta$. In the revised version, we show the performance of our approach at different mixing ratio $\lambda$ and provide a sensitivity analysis for all hyper-parameters. The experimental results are shown in Figure 6 in page 9, and Tables 9 and 10 in Appendix in page 16.
> |method/$\lambda$ ratio|0.2|1.0|2.0|5.0|10.0|
> |---|:--:|:--:|:--:|:--:|:--:|
> |ResNet-18|82.27|82.32|81.73|80.02|81.05|
> |ResNeXt-50|84.22|84.40|83.99|84.31|83.63|
>
> |method/N samples|N=1|N=2|N=3|N=4|N=5|
> |---|:--:|:--:|:--:|:--:|:--:|
> |Top1-Acc|78.04|82.16|82.32|81.78|80.79|
> |Top5-Acc|94.60|95.88|95.92|95.68|95.80|

---

> > ### Comment · Reviewer_vYYM · 2023-11-19
> > **Feedback to Authors' Rebuttal**
> >
> > Thanks for the detailed rebuttal, and the weaknesses and concerns I mentioned have been addressed. The updated comparison experiments on ImageNet and empirical analysis (e.g., occlusion robustness and gradCAM visualizations) have shown the effectiveness of the proposed modifications in AdAutoMix. Overall, I appreciate the proposed AdAutoMix from the perspective of adversarial training and decided to raise my score to 8. Moreover, I encourage the authors to further polish the writing issues (e.g., arrangement of tables and figures) and discuss the limitations & future works in the manuscript to ensure completeness.

---

> > > ### Author Response · Authors · 2023-11-23
> > >
> > > Dear Reviewer,
> > >
> > > Thank you for your very detailed suggestions to make our work fuller and more fulfilling. We apologise that due to the 9 page limit we were unable to include our future work and improvements within the paper.
> > > **In our future work, we will focus more on how to improve AdAutoMix so that it can have very little time overhead. And we also consider to further improve the " label mismatching " problem in some Mixup methods.**
> > > Finally, thank you very much for your recognition of our work, and we will keep up the good work, and we have a long way to go on the road of research.
> > >
> > > Best regards,
> > > Authors.

---

### Meta-Review · Area_Chair_42x3 · 2023-12-05

**Metareview:**

The paper proposes adding an adversarial component to Automixup, alternatively training the generator and target module for different losses. This encourages the generator to produce more diverse and mixed samples, which in turns makes the target module more robust. The authors preserve the original image prior by introducing a teacher and cosine similarity for the generator. The authors demonstrate that these techniques by achieving strong performance on seven classification benchmarks (CIFAR-100, Tiny-ImageNet, ImageNet-1K, CUB-200, FGBVC-Aircraft, and Stanford-Cars) and show that their technique outperforms existing all baselines given their network selection (ResNet18, ResNeXt50). They also show that their results improves the network's robustness. The paper does not have any glaring weaknesses. It would be nice if there were more ablations, but that's always true.

**Justification For Why Not Higher Score:**

It's not entirely clear how well the paper generalizes past classification (a task which is decreasing in importance in the vision community). In my opinion, demonstrating strong performance on another vision task would have been preferred for more than Spotlight.

**Justification For Why Not Lower Score:**

The paper shows strong performance on the baseline and adds an interesting and useful change to the Automixup formulation, which is currently one of the better augmentation techniques.

---

### Decision · Program_Chairs · 2024-01-16

Accept (spotlight)